# Dual Peptide-Modified Nanoparticles Improve Combination Chemotherapy of Etoposide and siPIK3CA Against Drug-Resistant Small Cell Lung Carcinoma

**DOI:** 10.3390/pharmaceutics12030254

**Published:** 2020-03-12

**Authors:** Hsin-Lin Huang, Wen Jen Lin

**Affiliations:** 1School of Pharmacy, College of Medicine, National Taiwan University, Taipei 10050, Taiwan; 2Drug Research Center, College of Medicine, National Taiwan University, Taipei 10050, Taiwan

**Keywords:** small cell lung carcinoma, drug-resistance, antagonist G peptide, etoposide, siPIK3CA

## Abstract

Small cell lung carcinoma (SCLC) is a highly aggressive form of malignancy with rapid recurrence and poor prognosis. The dual peptide-modified nanoparticles (NPs) for improving chemotherapy against drug-resistant small cell lung carcinoma cells has been developed. In this study, the SCLC targeting ligand, antagonist G peptide (AG), and cell-penetrating peptide, TAT, modified NPs were used to encapsulate both anticancer drugs etoposide (ETP) and PIK3CA small-interfering RNA (siPIK3CA). The ETP@NPs and siRNA@NPs had particle size 201.0 ± 1.9–206.5 ± 0.7 nm and 155.3 ± 12.4–169.1 ± 11.2 nm, respectively. The lyophilized ETP@NPs and siRNA@NPs maintained their particle size and zeta potential during 28-day storage without severe aggregation or dissociation. Either ETP@NPs or siRNA@NPs significantly reduced the IC_50_ of drugs by 2.5–5.5 folds and 2.4–3.9 folds, respectively, as compared to free ETP and siRNA/PEI nanocomplex in drug-resistant CD133(+) H69 cells. Herein, the IC_50_ of dual-peptide modified ETP@NPs and siRNA@NPs were prominently lower than single-peptide modified NPs. The synergistic effect (CI < 1) was further observed in co-treatment of ETP and siPIK3CA particularly delivered by dual-peptide modified NPs.

## 1. Introduction

It was revealed that tracheal, bronchial, and lung cancers result in high mortality. More than 90% of small cell lung carcinoma (SCLC) cases are attributed to smoking [1]. SCLC is a highly aggressive form of malignancy with rapid recurrence and poor prognosis, accounting for 11~20% of all lung cancers resulting in less than 5% 5-year survival rate [2,3]. Nowadays first-line combination chemotherapy for SCLC is etoposide (ETP) and platinum-based drugs [4]. Patients respond well to the chemotherapy at the beginning, but tumors relapse within one year because of drug resistance [5]. 

Antagonist G (AG; [Arg6, D-Trp7,9, NmePhe8]-substance P (6–11)) is a synthesized neuropeptide antagonist and can block several neuropeptides, such as GRP, vasopressin, bradykinin, and endothelin. AG peptide has been proved having inhibition activity on SCLC cell growth via block GRP receptor particularly in drug-resistant cells through in vitro cell and in vivo animal studies [6]. It was reported that AG-peptide can bind G-protein-coupled receptor (GPCR) and possess targeting ability [7]. AG has been used to modify the liposome surface where the AG-modified liposomes showed 20–40 folds increase of SCLC cellular association ability in H82 and H69 cells and enhanced the tumor growth inhibition of liposomal doxorubicin in tumor-bearing mice [8,9]. On the other hand, TAT (GRKKRRQRRR) is a cationic peptide with transmembrane ability [10,11]. Its transmembrane mechanism is concentration-dependent [12,13]. The low specificity is its hurdle in biomedical applications which can be improved by combination with peptide having targeting function [14]. 

ETP is the first choice for SCLC medication via deactivating DNA topoisomerase II in nuclei. It was reported that the CD133 protein is correlated to ETP resistance in cancer therapy where the CD133(+) cancer stem cells (CSCs) were increased in lung cancer cell line after treated by ETP [15,16]. Meanwhile, the G-protein-coupled receptor (GPCR) expression level in CD133(+) SCLC cells was observed higher than in CD133(−) cells [17]. PIK3CA small interfering RNA (siPIK3CA) can selectively inhibit PIK3CA gene expression to block SCLC cell proliferation and inhibit tumor growth [18,19]. Its specificity reduces the side effects induced by small molecular inhibitors in PI3K/AKT/mTOR transduction pathway. Polymer-based (e.g., hyaluronic acid, chitosan) and lipid-based nanoparticles have been applied for siRNA delivery and shown promising potential for gene therapy [20,21,22]. Recently, application of combinational therapeutic agents for cancer therapy have been designed for enhancing therapeutic efficacy and reducing side effect during chemotherapy [23,24,25,26,27]. 

It is important to explore a novel therapeutic strategy against drug-resistance during chemotherapy for SCLC. The strategy applied in this study was to develop dual-peptide modified nanoparticles with GPC receptor targeting and transmembrane functions to improve ETP chemotherapy with the aid of siPIK3CA in drug-resistant SCLC cells. In this study, the US FDA-approved poly(lactide-co-glycolide) (PLGA) was applied as the main polymer that was pegylated by poly(ethylene glycol) (PEG) to improve its in vivo circulation time as usual. The proposed SCLC targeting ligand, AG-peptide, and cell penetrating peptide, TAT, were used to modify the pegylated PLGA for NPs preparation where both anticancer drugs, ETP, and siPIK3CA, were encapsulated. The inhibition of cancer cell growth via combination therapy of ETP with the aid of siPIK3CA delivered by single as well as dual functional peptides-modified nanoparticles was investigated in CD133(+) H69 cells serving as an ETP-resistant SCLC cell model.

## 2. Materials and Methods

### 2.1. Materials

1-(3-Dimethylaminopropyl)-3-ethylcarbodiimide hydrochloride (EDC), *N*-ethyldiisopropylamine (*N*,*N*-Diisopropylethlamine) (DIEA, 99%), and thiazolyl blue tetrazolium bromide (MTT, 98%) were from Alfa Aesar (Echo Chemical Co., Ltd., Heysham, UK). *N*-Hydroxysuccinimide (NHS, 98%) and poly(vinyl alcohol) (PVA, 88% hydrolyzed, Mw 20,000~30,000 Da) were from Acros Organics Co. Inc. (NJ, USA). Poly(ethylene glycol) bis(amine) (PEG-diamine, Mw 5000 Da) was from Hunan Hua Teng Pharmaceutical Co., Ltd. (Hunan, China). Poly(D,L-lactide-co-glycolide) (PLGA, Mw~52,000 Da) was from Evonik Industries (Birmingham, AL, USA). Polyethylenimine (PEI, Mw~25,000 Da) was from Sigma-Aldrich Co., Ltd. (St. Louis, MO, USA). Etoposide (ETP) was from Hunan Hua Teng Pharmaceutical Co., Ltd. (Hunan, China). siPIK3CA (14,212.3 Da, GGUUAAAGAUCCAGAAGUAUU) was from DharmaconTM Inc. (Lafayette, CO, USA). Antagonist G-FITC peptide (1454.75 Da, FITC-Acp-Arg-(D-Trp)-N-Me-Phe-D-Trp-Leu-Met-OH) and TAT-TAMRA peptide (1809.13 Da, 5-TAMRA-Gly-Arg-Lys-Lys-Arg-Arg-Gln-Arg-Arg-Arg-OH) were from Kelowna International Scientific Inc. (Taipei, Taiwan). H69 human SCLC cell line was from American Type Culture Collection (Manassas, VA, USA). A549 human non-small cell lung carcinoma cell line was from Bioresource Collection and Research Center (Hsinchu, Taiwan). Dulbecco’s modified Eagle medium powder was from gibco by life technologies Co. (Grand Island, NY, USA). Anti-Human CD133 APC and Mouse IgG1 K Isotype Control APC were from eBioscience, Inc. (Wien, Österreich).

### 2.2. Synthesis of Peptide-Conjugated Copolymers

The peptide-conjugated copolymers including PLGA-PEG-AG and PLGA-PEG-TAT were synthesized based on previous method with modification [28,29]. Briefly, AG-peptide was reacted with PLGA-PEG in the presence of NHS and EDC (molar ratio 1:1:8:8) in dimethylformamide at room temperature for 24 h in dark. For PLGA-PEG-TAT synthesis, the molar ratio of TAT-peptide:PLGA-PEG:NHS:EDC was 1:1.5:7:7. The synthesized PLGA-PEG-AG and PLGA-PEG-TAT were precipitated with diethyl ether and centrifuged under 14,000 rpm at 4 °C for 10 min. The mixture was collected and washed with methanol three times followed by drying in desiccator under vacuum. The yield, the molecular weight, and the peptide conjugation ratio were determined.

### 2.3. Preparation and Characterization of ETP@NPs and siRNA@NPs

Four kinds of polymers including peptide-free PLGA-PEG, single peptide-conjugated PLGA-PEG-AG as well as PLGA-PEG-TAT, and dual peptide-conjugated PLGA-PEG-A/T polymers were used to prepare NPs for encapsulation of ETP and siPIK3CA. The ETP loaded NPs (ETP@NPs) were prepared by using a single emulsion solvent evaporation method [30]. ETP and each copolymer (1.67:5 weight ratio) were weighed and dissolved in dichloromethane/acetone (9:1 *v*/*v*). The ETP-polymer solution was added into 0.5% polyvinyl alcohol solution (O/W 1:4 *v*/*v*) under sonication in an ice bath for 5 min followed by magnetic stir for 4 h. The siRNA-loaded NPs (siRNA@NPs) were prepared by double emulsion solvent evaporation method. The siRNA/PEI complex was prepared first where siRNA and PEI were dissolved in TE buffer and mixed for 30 s followed by letting it stand for 30 min in dark. The siRNA/PEI complex was added into four kinds of polymer solutions to form w/o emulsion which was further added into 2.5% polyvinyl alcohol solution (w/o/w volume ratio 0.1:1:6) under sonication in an ice bath for 3 min followed by magnetic stir for 4 h. The remaining organic solvent was removed by rotary evaporator under reduced pressure at 35 °C. The ETP@NPs and siRNA@NPs were collected after centrifugation and washed with water twice. Finally, the NPs were frozen at −80 °C followed by freeze drying (EZ-550R, FTS Systems, Stone Ridge, NY, USA). The yields of prepared NPs were calculated. The particle sizes and zeta potential were determined by zetasizer (Nano-ZS 90 Zetasizer, Malvern Instruments, Worcestershire, UK). The amount of ETP encapsulated by NPs was determined by HPLC at 240 nm, and of siRNA was determined by using fluorescence spectrometer at excitation wavelength 633 nm and emission wavelength 670 nm. The drug loading and encapsulation efficiency were calculated. The stability of ETP@NPs and siRNA@NPs was monitored after lyophilization. At day 0, 7, 14, 21, and 28 after storage, the samples were collected and redispersed in deionized water. The particle size and zeta potential of each sample were measured. 

### 2.4. Sorting of CD133(+) H69 Cells

Total of 2 × 10^5^ cells were dispersed in 100 μL of FACS buffer in each of two tubes. One was as the negative control. The isotype IgG-APC Ab 2 μL was added into another tube as the positive control. PBS 3 mL was then added and centrifuged at 1300 rpm for 5 min. The cells were redispersed in 600-μL sorting buffer. For testing group, 10^6^ cells were dispersed in 100 μL of FACS buffer in the tube, and CD133-APC Ab 2 μL was added into the cells. PBS 50 mL was then added and centrifuged at 1300 rpm for 5 min. The supernatant was removed and the cells were redispersed in 2 mL sorting buffer and then sorted by flow cytometer. The CD133(+) H69 cells were sorted from the top 10% of the parent H69 cells, and the CD133 protein expression level was determined. A total of 10,000 events were analyzed, and the mean fluorescence intensity (MFI) was recorded by BD CellQuest Pro software (version 6.0, San Jose, CA, USA). The morphology of CD133(+) H69 cells was observed by inverted microscope (Olympus CKX41, Tokyo, Japan). The sorted CD133(+) H69 cells were served as ETP-resistant SCLC cell model for following studies.

### 2.5. Cellular Uptake

The CD133(+) H69 cells, a high-expressed GPCR cell line, were uniformly seeded in 24-well plates at a density of 2 × 10^5^ cells/well in SITA and incubated for 24 h. The Cy5 loaded PLGA-PEG NPs, PLGA-PEG-AG NPs, PLGA-PEG-TAT NPs, and PLGA-PEG-A/T NPs were added and incubated in 5% CO_2_ at 37 °C for 2 h. The cells were washed with 3 mL PBS followed by centrifugation at 1300 rpm for 5 min two times. On the other hand, a GPCR low-expressed A549 cell line was uniformly seeded in 6-well plates at a density of 2 × 10^5^ cells/well in DMEM containing 10% FBS and incubated for 24 h. The supernatant was removed, and the cells were washed with PBS. Four kinds of Cy5 loaded NPs in free DMEM were added and incubated in 5% CO_2_ at 37 °C for 2 h. The cells were washed with 3 mL PBS followed by centrifugation at 1300 rpm for 5 min two times. The cells were washed with PBS and trypsinized followed by centrifugation repeatedly. Finally, both cells were collected and the fluorescence intensity was measured by flow cytometer (BD FACSCalibur Becton Dickinson, Franklin Lakes, NJ, USA). A total of 10,000 events were analyzed for each sample, and the mean fluorescence intensity (MFI) was recorded.

### 2.6. Endocytosis Mechanism

The endocytosis pathway of PLGA-PEG-AG NPs, PLGA-PEG-TAT NPs, and PLGA-PEG-A/T NPs were investigated in CD133(+) H69 cells. The cells were pre-treated with endocytosis inhibitors including chlorpromazine (10 μg/mL, clathrin inhibitor), nystatin (50 μg/mL, caveolin inhibitor), and amiloride (230 μg/mL, macropinocytosis inhibitor), respectively, for 2 h. After that, NPs were added and incubated for additional 2 h. The cells were washed with PBS and centrifuged repeatedly. Finally, the cells were collected and the fluorescence intensity was measured by flow cytometer. The inhibition efficiency (%) was calculated based on the MFI obtained from inhibitor-treated group relative to control group without inhibitor treatment. 

### 2.7. Cytotoxicity Study

The cell viability of nanocarriers including PLGA-PEG NPs, PLGA-PEG-AG NPs, PLGA-PEG-TAT NPs, and PLGA-PEG-A/T NPs was performed in both CD133(+) H69 and A549 cells. CD133(+) H69 cells were uniformly seeded in 24-well plates at a density of 5 × 10^4^ cells/well in SITA, and A549 cells were seeded in at a density of 2 × 10^4^ cells/well in DMEM containing 10% FBS. After 24-h incubation, the medium was discarded and various concentrations of NPs (50–800 μg/mL) were added. The cells were further incubated in 5% CO_2_ at 37 °C for 24 h. Then, the MTT solution was added to each well and continuously incubated for additional 4 h. The supernatant was removed and dimethyl sulfoxide (DMSO) was added to dissolve the formazan crystal. The absorbance was measured at 570 nm and 690 nm using a microplate reader, and the cell viability was calculated. The cells without NPs treatment served as the control group. On the other hand, the cytotoxicity of drug loaded ETP@NPs (5–150 μg/mL) and siRNA@NPs (50–1000 ng/mL) was further performed in CD133(+) H69 cells following the procedure mentioned above. The IC_50_ (half-maximal inhibitory concentration) was calculated by fitting the semi-log plot of cell viability (%) versus logarithm of drug concentrations using SigmaPlot 12.5 software (Systat Software, Inc., CA, USA). In combination cytotoxicity study, the ETP@NPs (5 and 40 μg/mL) and siRNA@NPs (100, 200, and 400 ng/mL) were co-treated with the cells, and the percentage of cell viability was determined. The combination index theorem combined with median-effect principle was applied to calculate the combination index (CI) by using CompuSyn (version 1, ComboSyn, Inc., Paramus, NJ, USA) based on the combination cytotoxicity data [31]. CI < 1 indicates the synergistic effect between ETP and siRNA, CI = 1 represents additive effect, and CI > 1 suggests the presence of antagonism between both.

### 2.8. Statistical Analysis

All data were presented as mean ± SD. All the statistics analysis was conducted by SigmaPlot 12.5 (Systat Software Inc., CA, USA). One-way ANOVA and independent Student’s *t*-test were used, and the statistical significance was defined as *p* < 0.05.

## 3. Results and Discussion

### 3.1. Characterization of Peptide-Conjugated Copolymers

The PLGA-PEG-AG and PLGA-PEG-TAT were synthesized and characterized. There were four indicating peaks including peak *a* at δ5.2 ppm for CH proton of lactide, peak *b* at δ4.8 ppm for CH2 proton of glycolide, peak *c* at δ1.5 ppm for CH3 proton of lactide, and peak *d* at δ3.6 ppm for CH2 proton of ethylene glycol. It confirmed the presence of both PLGA and PEG domains in the PLGA-PEG copolymer. The yield of PLGA-PEG-AG copolymer was 76.5 ± 0.8%. The weight-average molecular weight (M_w_), number-average molecular weight (M_n_), and polydispersity (PD) were 67,000 ± 5000 Da, 42,000 ± 2000 Da, and 1.61 ± 0.06, respectively. The AG-peptide conjugation ratio was 102.7 ± 3.6 mol%. For PLGA-PEG-TAT copolymer, the yield was 86.0 ± 1.7%, and the M_w_, M_n_, as well as PD were 76,000 ± 1,000 Da, 52,000 ± 3000 Da, and 1.45 ± 0.09, respectively. The TAT-peptide conjugation ratio was 91.8 ± 8.2 mol%.

### 3.2. Characterization of Peptide-Conjugated Nanoparticles

There were four kinds of NPs prepared and investigated in this study including unmodified PLGA-PEG NPs, single peptide-modified PLGA-PEG-AG NPs as well as PLGA-PEG-TAT NPs, and dual peptide-modified PLGA-PEG-A/T NPs. The particle size, polydispersity index (PDI), zeta potential, and yields of these NPs are shown in Table 1. All NPs had yields higher than 75%. The size of NPs was in the range of 166.9 ± 5.3–174.0 ± 8.4 nm with mono-sized distribution (PDI < 0.2). There was no prominent size change in NPs because of the presence of peptides. The zeta potentials of PLGA-PEG NPs, PLGA-PEG-AG NPs, PLGA-PEG-TAT NPs, and PLGA-PEG-A/T NPs were 26.7 ± 1.8, 23.1 ± 4.3, 35.3 ± 2.2, and 24.1 ± 0.5 mV, respectively, showing an increasing positive charge manner with the addition of arginine-rich TAT-peptide. Figure 1 shows the TEM images of these NPs. All NPs were well separated with homogeneous size distribution, and there was no significant change in morphology of NPs after conjugation of peptides. 

### 3.3. Characterization of ETP@NPs and siRNA@NPs

ETP and siRNA were encapsulated by peptide-free PLGA-PEG NPs, single-peptide PLGA-PEG-AG NPs as well as PLGA-PEG-TAT NPs, and dual-peptide PLGA-PEG-A/T NPs by solvent evaporation method. The characteristics of these drug-loaded ETP@NPs and siRNA@NPs are listed in Table 2 and Table 3, respectively. The yields of ETP@NPs were 86.1 ± 1.9-91.7 ± 5.0%. The particle size was in the range of 201.0 ± 1.9–206.5 ± 0.7 nm with PDI < 0.2 indicating monodispersity character. All ETP@NPs had positive zeta potential in the range of 36.3 ± 2.4–39.6 ± 1.0 mV where TAT-NPs-ETP NPs exhibited the most positive charge because of the conjugation of arginine-rich TAT-peptide. The encapsulation efficiency of ETP was higher than 60%, and the drug loading was 10.1 ± 0.8–12.3 ± 1.8%. Basically, the physicochemical properties of these NPs were similar regardless of the presence of peptides. On the other hand, the yields of siRNA@NPs were 54.5 ± 2.9–64.9 ± 2.0%. The particle size was in the range of 155.3 ± 12.4–169.1 ± 11.2 nm with PDI < 0.2. All siRNA@NPs possessed positive-charged character (31.0 ± 2.7–34.1 ± 1.4 mV). The encapsulation efficiency of siRNA was 58.6 ± 5.0–71.6 ± 15.6%, and the drug loading was 0.24 ± 0.03–0.29 ± 0.05%.

The peptide-free and peptide-conjugated NPs before encapsulation of ETP and siRNA had mean particle size in the range of 166.9–174.0 nm. The particle size of ETP@NPs after encapsulation of ETP was enlarged to 201.0–206.5 nm because of the loading of ETP in the hydrophobic core of the NPs. However, the mean size of siRNA@NPs (155.3–169.1 nm) was even smaller than that of the blank NPs. It seems that PEI and siRNA formed compact nanocomplex inside siRNA@NPs which shrank the NPs size. The similar result has been reported where the particle size of CaP-siRNA-PLGA-PEI NPs in the presence of PEI was smaller than that of CaP-siRNA-PLGA NPs [32]. The surface zeta potential of NPs showed varied dependence of the peptides. The peptide-free PLGA-PEG NPs possessed positive charge because of the exposure of amino end groups in PEG block. Herein, the TAT-peptide-bearing positive-charged arginine contributed the most positive charge of PLGA-PEG-TAT. Many studies reported that the NPs with size < 200 nm and carrying positive charge are feasible for tumor accumulation via the enhanced permeability and retention (EPR) effect [33,34]. Our ETP@NPs and siRNA@NPs had particle size ~200 nm with positive charge character, which meets the criteria, and may possess the potential to accumulate in the tumor cells. 

### 3.4. Stability of ETP@NPs and siRNA@NPs

The stability of ETP@NPs and siRNA@NPs after lyophilization was monitored by their particle size and zeta potential change for 28 days, and the result is shown in Figure 2. The ETP@NPs maintained their particle size within the range of 80–120% relative to the freshly prepared NPs, and the PDI values remained <0.23. On the other hand, siRNA@NPs maintained their particle size within the range of 90–130% relative to the freshly prepared NPs, and the PDI values were <0.20. Both ETP@NPs and siRNA@NPs maintained their zeta potential during storage, too. All of these results illustrated the lyophilized ETP@NPs and siRNA@NPs maintained their stability during 28-day storage without severe aggregation or dissociation. It seems that the hydrophilic PEG plays an important role in maintaining NPs stability via the surrounding on the outer shell of the NPs. 

### 3.5. Cellular Uptake

The CD133(+) H69 cells were sorted from the top 10% of the parent H69 cells, and the CD133 protein expression level was high up to 96.7% (Figure 3A). The morphology of CD133(+) H69 cells appeared as floating aggregates as reported (Figure 3B) (ATCC official website, 2019). The ETP-resistance character of CD133(+) H69 SCLC cell model was further confirmed where CD133(+) H69 and parent H69 cells were treated with ETP in the concentrations of 5-200 μg/mL for 72 h. The cell viability of CD133(+) H69 showed significantly higher than that of parent H69 (*p* < 0.001), and the corresponding IC_50_ values were 81.9 ± 5.8 μg/mL and 32.8 ± 3.1 μg/mL, respectively (Figure 3C). The CD133(+) H69 cells sorted from top 10% of the parent H69 cells have high CD133 protein expression level and ETP-resistant character. Therefore, CD133(+) H69 was served as a GPCR high-expressed ETP-resistant SCLC cell model in this study, whereas A549 was selected as a GPCR low-expressed non-resistant cell line as reported [35]. 

Figure 4 shows the cellular uptake of PLGA-PEG NPs, PLGA-PEG-AG NPs, PLGA-PEG-TAT NPs, and PLGA-PEG-A/T NPs in CD133(+) H69 and A549 cells for 2 h. The cellular uptake in CD133(+) H69 cells was in the order of PLGA-PEG-TAT NPs > PLGA-PEG-A/T NPs > PLGA-PEG-AG NPs > PLGA-PEG NPs. PLGA-PEG-AG NPs had cellular uptake significantly different from PLGA-PEG NPs in CD133(+) H69 cells (*p* < 0.001) but similar to each other in A549 cells (73.7 ± 1.8% vs. 69.4 ± 6.9%). It implied that AG peptide has the potential to enhance cellular uptake of NPs in GPCR high-expressed drug-resistant CD133(+) H69 SCLC cells. On the other hand, TAT-peptide modified PLGA-PEG-TAT NPs and PLGA-PEG-A/T NPs expressed even higher cellular uptake than PLGA-PEG-AG NPs in both CD133(+) H69 and A549 cells indicating non-specificity of transmembrane TAT-peptide. The similar cellular uptake tendency was observed from fluorescence microscopic images of CD133(+) H69 cells shown in Figure 4C. All of these results confirmed that AG-peptide had the potential as a targeting ligand to enhance cellular uptake of NPs in GPCR high-expressed drug-resistant CD133(+) H69 SCLC cells. The similar result has been reported for AG-liposomes [9]. The transmembrane property of TAT-peptide exerted its benefit to enhance cellular uptake of NPs which further overcame the multi-drug resistance that occurred in CD133(+) H69 cells. Gullotti et al. found that the paclitaxel (PTX) loaded PLGA-TAT NPs is more effective than free PTX in multi-drug resistant NCI/ADR-RES cancer cells [36].

### 3.6. Endocytosis Mechanism

Figure 5 illustrates the cellular uptake of PLGA-PEG-AG NPs, PLGA-PEG-TAT NPs, and PLGA-PEG-A/T NPs in CD133(+) H69 cells after pre-treated by chlorpromazine (clathrin-mediated endocytosis inhibitor), nystatin (caveolin-mediated endocytosis inhibitor), and amiloride (macropinocytosis inhibitor), respectively. The result revealed that the cellular uptake of PLGA-PEG-AG NPs was through clathrin-mediated endocytosis and macropinocytosis, and of PLGA-PEG-TAT NPs was dominated by caveolin-mediated endocytosis. Nevertheless, the cellular uptake of dual-peptide PLGA-PEG-A/T NPs combined clathrin- with caveolin-mediated endocytosis. Dual-peptide modification allowed NPs to efficiently enter into the CD133(+) H69 cells via multiple endocytosis pathways [37].

### 3.7. Cytotoxicity of Peptide-Free and Peptide-Conjugated NPs

The cell viability of PLGA-PEG NPs, PLGA-PEG-AG NPs, PLGA-PEG-TAT NPs, and PLGA-PEG-A/T NPs was evaluated in CD133(+) H69 and A549 cell lines for 72 h, and the result is shown in Figure 6. The concentrations of NPs corresponding for >80% viability of CD133(+) H69 cells were in order of PLGA-PEG-TAT NPs (<50 μg/mL) < PLGA-PEG-AG NP (<200 μg/mL) < PLGA-PEG NPs ~ PLGA-PEG-A/T NPs (< 400 μg/mL), and of A549 cells were PLGA-PEG-TAT NPs ~ PLGA-PEG-A/T NP (<100 μg/mL) < PLGA-PEG NPs ~ PLGA-PEG-AG NPs (<800 μg/mL). The PLGA-PEG-TAT NPs expressed the lowest cell viability than the others, no matter in CD133(+) H69 or CD133(−) A549 cancer cells. The positive-charge bearing arginine-rich TAT-peptide accounted for higher cytotoxicity of PLGA-PEG-TAT NPs than the others. However, replacing partial TAT-peptide by AG-peptide effectively reduced the cytotoxicity of TAT-peptide-modified NPs in CD133(+) H69 cells where the tolerance concentration of dual-peptide PLGA-PEG-A/T NPs for >80% cell viability was elevated to 400 μg/mL as compared to the PLGA-PEG-TAT NPs (<50 μg/mL).

### 3.8. Cytotoxicity of ETP@NPs and siRNA@NPs

The cytotoxicity of drug-loaded ETP@NPs and siRNA@NPs was further investigated in CD133(+) H69 cells for 72 h, and the results are shown in Figure 7. The corresponding IC_50_ of free ETP and ETP@NPs was in order of A/T-NPs-ETP (15.0 ± 2.9 μg/mL) < TAT-NPs-ETP (24.2 ± 0.8 μg/mL) ~ AG-NPs-ETP (26.4 ± 2.2 μg/mL) < NPs-ETP (32.2 ± 0.7 μg/mL) < free ETP (81.9 ± 5.8 μg/mL) (Figure 7A). In other words, the IC_50_ of A/T-NPs-ETP, TAT-NPs-ETP, AG-NPs-ETP, and NPs-ETP was 5.5, 3.4, 3.1, and 2.5-fold reduced relative to free ETP. Both AG- and TAT-modified NPs apparently showed effective inhibition of cell viability, and the dual-peptide A/T-NPs-ETP produced the highest cytotoxicity in terms of the lowest IC_50_. Figure 7B illustrates the cytotoxicity in CD133(+) H69 cells after treating with siRNA/PEI nanocomplex and siRNA@NPs for 72h. All siRNA@NPs-induced higher cytotoxicity than siRNA/PEI nanocomplex, and the IC_50_ was in order of A/T-NPs-siRNA (169.7 ± 17.1 ng/mL) < AG-NPs-siRNA (203.7 ± 2.1 ng/mL) < TAT-NPs-siRNA (258.6 ± 17.0 ng/mL) < NPs-siRNA (280.7 ± 4.4 ng/mL) < siRNA/PEI nanocomplex (665.4 ± 6.7 ng/mL). It seems that the siRNA/PEI complex did not provide enough protection to avoid enzymatic degradation of siRNA, where the cytotoxicity in terms of IC_50_ of A/T-NPs-siRNA, TAT-NPs-siRNA, AG-NPs-siRNA, and NPs-siRNA was 3.9, 3.3, 2.6, and 2.4-fold reduced relative to siRNA/PEI nanocomplex. Herein, the dual-peptide A/T-NPs-siRNA exhibited the highest cytotoxicity with the lowest IC_50_ in CD133(+) H69 cells as A/T-NPs-ETP did. In other words, the dual-peptide modified A/T-NPs-ETP as well as A/T-NPs-siRNA exhibited the lowest IC_50_ even better than ETP and siRNA loaded single-peptide-modified AG-NPs and TAT-NPs. All of these results indicated that AG-NPs combined with TAT-NPs could efficiently promote NPs endocytosis into ETP-resistant cells to inhibit cell growth. 

In addition to the peptides as the potential ligands to efficiently promote NPs endocytosis into cancer cells and inhibit cell growth, the alternative methods can also be useful, such as using macromolecules as the targeting ligand and/or applying magnetic hyperthermia treatment. Das et al. designed folate nanoconjugates which not only induced remarkable cytotoxicity but also significantly suppressed tumor growth in vivo [38]. The similar anti-cancer effect was observed by using thermosensitive doxorubicin-loaded magnetoliposomes [39]. The in vivo results showed a prominent antiglioma efficacy because of the magnetic drug targeting combined with magnetic hyperthermia effect of superparamagnetic iron oxide NPs.

### 3.9. Combination Treatment of ETP@NPs and siRNA@NPs

Figure 8 illustrates the combination cytotoxicity after co-treatment of ETP (5, 40 μg/mL) and siRNA (100, 200, 400 ng/mL) loaded peptide-free, single-peptide, and dual-peptide modified ETP@NPs and siRNA@NPs in CD133(+) H69 cells for 72 h. All peptide-modified NPs (Figure 8B–8D) induced higher cytotoxicity than peptide-free NPs (Figure 8A) in order of A/T-NPs > AG-NPs > TAT-NPs > NPs after co-treatment of both ETP@NPs and siRNA@NPs. This combination cytotoxicity exhibited an increasing tendency as raising treatment dose of ETP@NPs and siRNA@NPs. Herein, the ETP@NPs containing 5–40 μg/mL of ETP combined with siRNA@NPs containing 100 ng/mL of siRNA achieved significantly better combination cytotoxicity than free ETP drug and ETP@NPs alone at the same ETP concentration. As shown in Figure 3C, 41.2 ± 0.7% of CD133(+) H69 cell survival was seen after treatment with high concentration of free ETP drug at 200 μg/mL. However, there were less than 26% of cell survival after co-treatment of low dose ETP@NPs (equivalent of 40 μg/mL ETP) combined with siRNA@NPs (equivalent of 100 ng/mL siRNA). This result implied that siRNA plays an important role in enhancing cytotoxicity of ETP. Furthermore, the combination index (CI) was calculated based on these combination cytotoxicity data, and the results are shown in the right hand side of Figure 8. The co-treatment of ETP@NPs and siRNA@NPs had CI < 1 at all combination concentrations no matter NPs with or without peptide modification. It confirmed that the combination therapy of ETP@NPs and siRNA@NPs successfully induced synergistic cytotoxicity in drug-resistant cells. Herein, the dual-peptide-modified A/T-NPs induced the most prominent combination of cytotoxicity effect to effectively eradicate ETP-resistant CD133(+) H69 cells with stem cell character which provided a promising potential to reduce cancer recurrence probability.

The applications of short chain peptides to improve therapeutic efficacy of chemotherapeutic agents delivered by nanocarriers have been wildly investigated for cancer therapy. Herein, the cationic peptides with cell-penetrating ability and the functional peptides with receptor-targeting capability are most attractive. In our study, the receptor-targeting AG-peptide and cell-penetrating TAT-peptide-conjugated NPs successfully induced synergistic cytotoxicity to overcome drug-resistance during chemotherapy for SCLC. Zhang et al. used cell-penetrating R9 peptide to conjugate polymeric nanocarrier composed of pegylated poly(hexyl ethylene phosphate) which was further coated by polyanionic polyphosphoester with tumor acidity-activatable property [40]. The result showed that R9 peptide-conjugated NPs significantly enhanced cellular uptake in 4T1 tumor cells and promoted anti-tumor efficiency in tumor-bearing mice. Using RGD targeting peptide decorated lipoprotein has been proved to improve the chemophotodynamic therapy of triple negative breast cancer [41]. The deep tumor-penetration and cancer cells-accessing abilities resulted in prominent inhibition of solid tumor growth. Recently, a novel AAN peptide has been found that can target the tumor-associated macrophages. The AAN peptide-modified doxorubicin loaded liposomes showed a prominent antitumor efficacy in 4T1 breast cancer-bearing mice [42]. All of these substantial findings further revealed the important roles of these functional peptides in nanomedicine-related cancer therapy.

## 4. Conclusions

The AG-peptide and TAT-peptide-conjugated copolymers were synthesized and applied to prepare NPs for encapsulation of ETP and siPIK3CA. The AG-peptide and TAT-peptide modified NPs through GRP receptor targeting and transmembrance ability promoted endocytosis of ETP and siPIK3CA into drug-resistant cells. The dual-peptide modified A/T-NPs-ETP and A/T-NPs-siRNA induced better cytotoxic effect with lower IC_50_ than drug-loaded single-peptide-modified AG-NPs and TAT-NPs. Combination therapy of ETP with the aid of siPIK3CA particularly delivered by dual peptide-modified NPs further exerted synergistic cytotoxicity in drug-resistant CD133(+) H69 SCLC cells. 

## Figures and Tables

**Figure 1 pharmaceutics-12-00254-f001:**
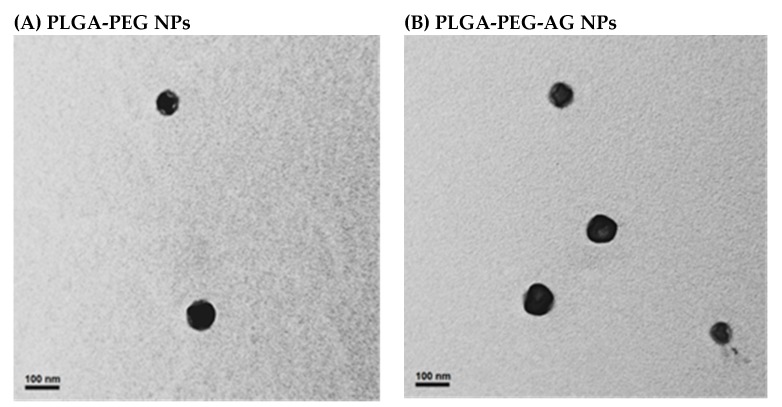
TEM photographs of (**A**) PLGA-PEG NPs, (**B**) PLGA-PEG-AG NPs, (**C**) PLGA-PEG-TAT NPs and (**D**) PLGA-PEG-A/T NPs (scale bar: 100 nm).

**Figure 2 pharmaceutics-12-00254-f002:**
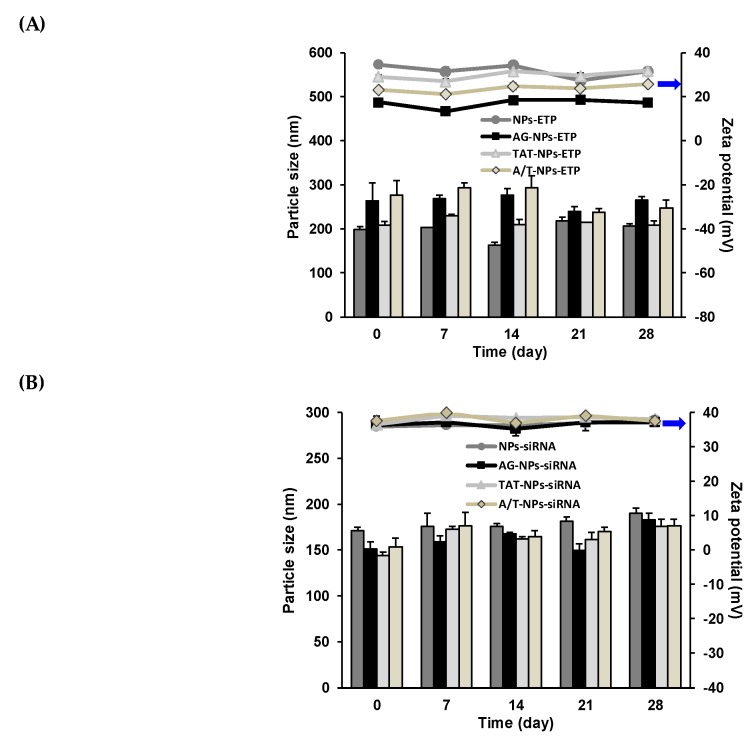
Stability of lyophilized (**A**) ETP@NPs and (**B**) siRNA@NPs at −20 °C for 28 days by monitoring the particle size (bar) and zeta potential (line) change during storage. (*n* = 3, mean ± SD).

**Figure 3 pharmaceutics-12-00254-f003:**
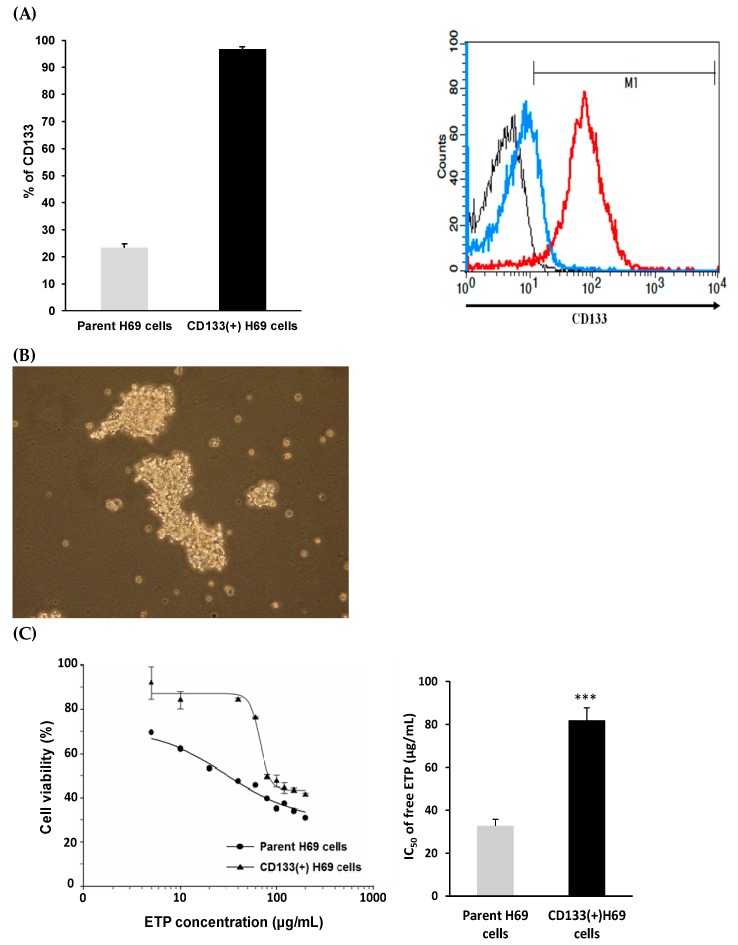
(**A**) CD133(+) H69 cells after sorting with high expression of CD133. (black: isotype IgG-APC negative control; blue: CD133 expression in parent H69 cells; red: CD133 expression in CD133(+) H69 cells. (**B**) Morphology of CD133(+) H69 cells (40×). (**C**) IC_50_ of ETP in CD133(+) H69 and parent H69 cells after treated by free ETP for 72h. (*** *p* < 0.001, *n* = 3).

**Figure 4 pharmaceutics-12-00254-f004:**
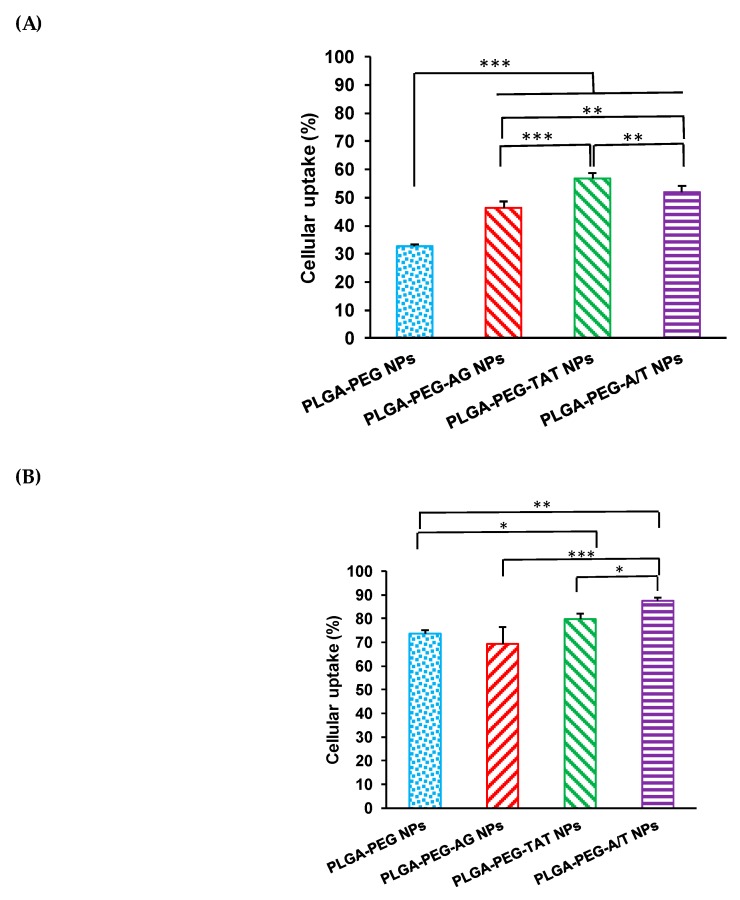
Flow cytometric analysis of cellular uptake of PLGA-PEG NPs, PLGA-PEG-AG NPs, PLGA-PEG-TAT NPs and PLGA-PEG-A/T NPs in (**A**) CD133(+) H69 and (**B**) A549 cell lines for 2 h (* *p* < 0.05, ** *p* < 0.01, *** *p* < 0.001, *n* = 3) and (**C**) fluorescence microscopic images of CD133(+) H69 cells treated with medium only and Cy5 loaded NPs (scale bar: 20 μm).

**Figure 5 pharmaceutics-12-00254-f005:**
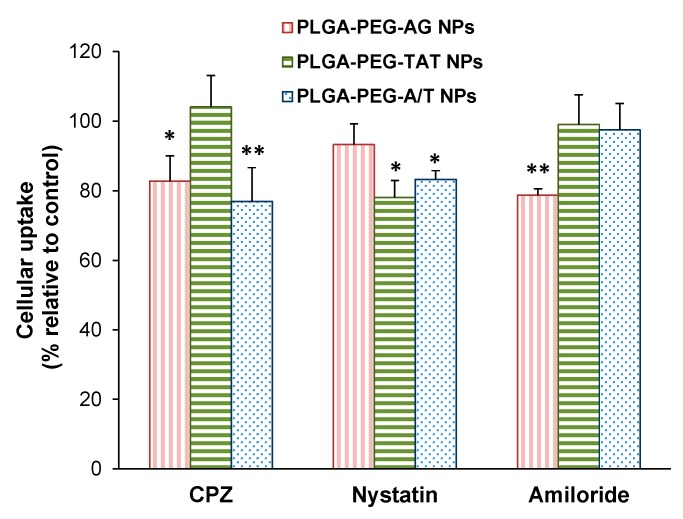
Effects of endocytosis inhibitors on cellular uptake of PLGA-PEG-AG NPs, PLGA-PEG-TAT NPs and PLGA-PEG-A/T NPs in CD133(+) H69 cells. (* *p* < 0.05, ** *p* < 0.01, compared to NPs without pre-treated by inhibitors, *n* = 3).

**Figure 6 pharmaceutics-12-00254-f006:**
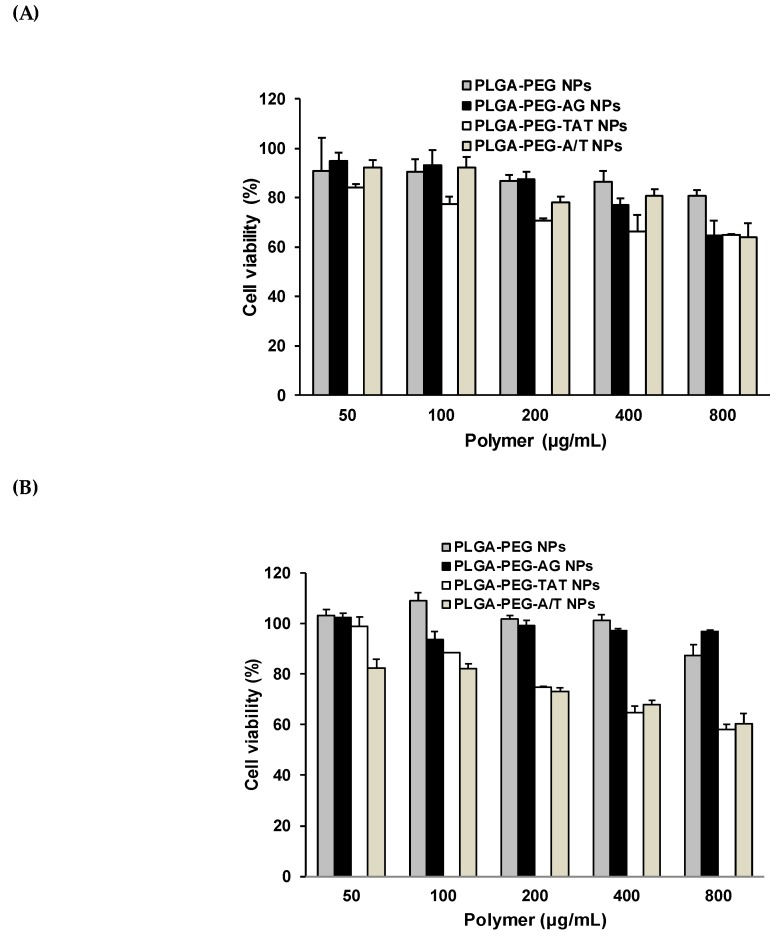
Cell viability of PLGA-PEG NPs, PLGA-PEG-AG NPs, PLGA-PEG-TAT NPs, and PLGA-PEG-A/T NPs in (**A**) CD133(+) H69 cells and (**B**) A549 cells for 72 h (*n* = 3).

**Figure 7 pharmaceutics-12-00254-f007:**
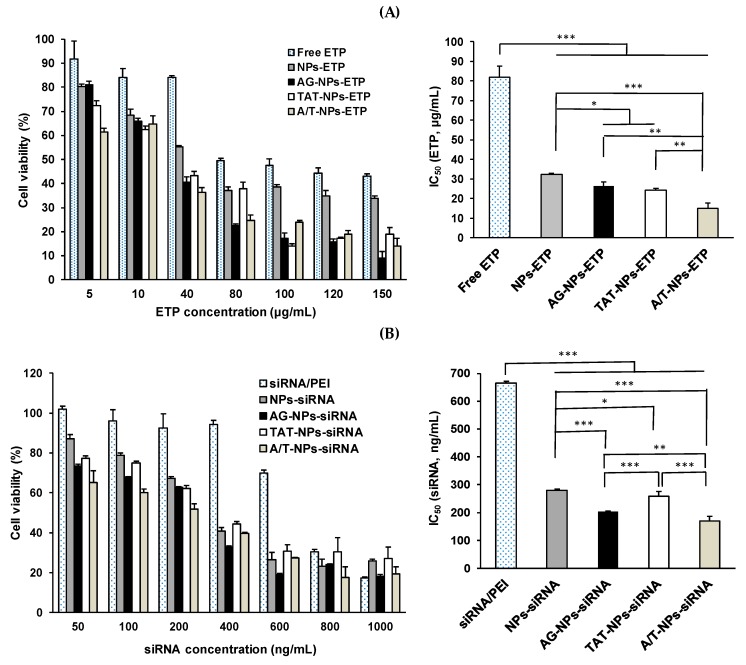
Cytotoxicity in CD133(+) H69 cells after treated by peptide-free, single-peptide and dual-peptide modified (**A**) ETP@NPs and (**B**) siRNA@NPs for 72h and their corresponding IC_50_. (* *p* < 0.05, ** *p* < 0.01, *** *p* < 0.001, *n* = 3).

**Figure 8 pharmaceutics-12-00254-f008:**
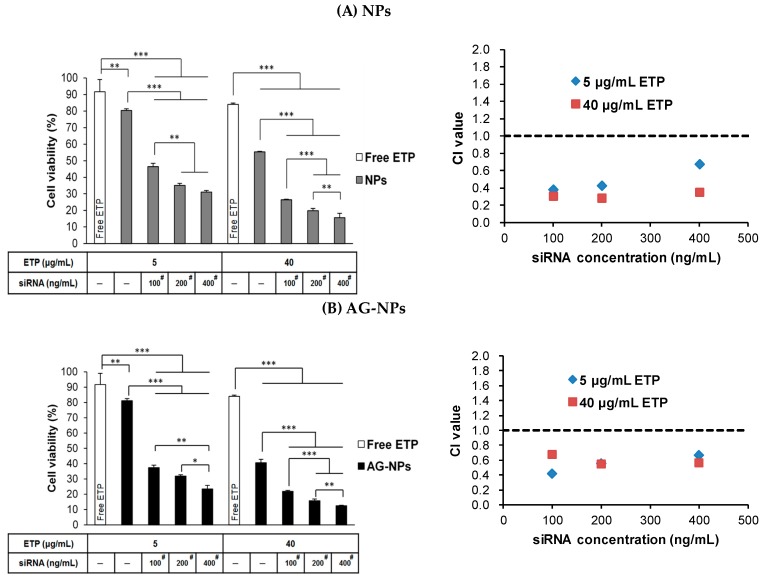
Combination cytotoxicity in CD133(+) H69 cells and their corresponding CI values after co-treatment of ETP@NPs and siRNA@NPs for 72 h. (**A**) NPs, (**B**) AG-NPs, (**C**) TAT-NPs, and (**D**) A/T-NPs. (* *p* < 0.05, ** *p* < 0.01, *** *p* < 0.001, *n* = 3). ^#^ significant difference compared to NPs-ETP, AG-NPs-ETP, TAT-NPs-ETP, or A/T-NPs-ETP alone.

**Table 1 pharmaceutics-12-00254-t001:** The particle sizes, polydispersity index (PDI), zeta potentials, and yields of PLGA-PEG NPs, PLGA-PEG-AG NPs, PLGA-PEG-TAT NPs, and PLGA-PEG-A/T NPs (*n* = 3).

NPs	PLGA-PEG	PLGA-PEG-AG	PLGA-PEG-TAT	PLGA-PEG-A/T
**Particle size (nm)**	166.9 ± 5.3	167.8 ± 9.8	174.0 ± 8.4	169.2 ± 7.6
**PDI**	0.15 ± 0.06	0.13 ± 0.05	0.13 ± 0.03	0.11 ± 0.01
**Zeta potential (mV)**	26.7 ± 1.8	23.1 ± 4.3	35.3 ± 2.2	24.1 ± 0.5
**Yield (%)**	75.3 ± 2.9	79.5 ± 10.3	85.3 ± 3.6	81.1 ± 2.5

**Table 2 pharmaceutics-12-00254-t002:** Characteristics of ETP@NPs (*n* = 3).

ETP@NPs	NPs-ETP	AG-NPs-ETP	TAT-NPs-ETP	A/T-NPs-ETP
**Particle size (nm)**	201.0 ± 1.9	201.3 ± 4.6	202.3 ± 3.5	206.5 ± 0.7
**PDI**	0.13 ± 0.01	0.08 ± 0.04	0.08 ± 0.07	0.05 ± 0.05
**Zeta potential (mV)**	36.3 ± 2.4	36.3 ± 1.0	39.6 ± 1.0	36.6 ± 1.5
**EE (%)**	61.6 ± 2.6	61.4 ± 12.9	61.1 ± 6.0	79.3 ± 15.3
**Drug loading (%)**	10.1 ± 0.8	10.2 ± 2.3	10.2 ± 1.2	12.3 ± 1.8
**Yield (%)**	87.2 ± 4.2	86.7 ± 1.7	86.1 ± 1.9	91.7 ± 5.0

**Table 3 pharmaceutics-12-00254-t003:** Characteristics of siRNA@NPs (*n* = 3).

siRNA@NPs	NPs-siRNA	AG-NPs-siRNA	TAT-NPs-siRNA	A/T-NPs-siRNA
**Particle size (nm)**	166.2 ± 6.2	162.6 ± 21.0	155.3 ± 12.4	169.1 ± 11.2
**PDI**	0.16 ± 0.06	0.13 ± 0.01	0.10 ± 0.03	0.12 ± 0.03
**Zeta potential (mV)**	34.1 ± 1.4	31.0 ± 2.7	33.2 ± 0.9	32.9 ± 2.3
**EE (%)**	60.7 ± 5.4	63.2 ± 7.4	58.6 ± 5.0	71.6 ± 15.6
**Drug loading (%)**	0.26 ± 0.01	0.24 ± 0.03	0.27 ± 0.03	0.29 ± 0.05
**Yield (%)**	57.0 ± 2.8	64.9 ± 2.0	54.5 ± 2.9	61.6 ± 3.1

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
