# Peer review of "Dual Peptide-Modified Nanoparticles Improve Combination Chemotherapy of Etoposide and siPIK3CA Against Drug-Resistant Small Cell Lung Carcinoma"

_pharmaceutics, 2020, doi:10.3390/pharmaceutics12030254_

Round 1

Reviewer 1 Report

The paper entitled "Dual Peptide-Modified Nanoparticles Improve Combination Chemotherapy of Etoposide and siPIK3CA against Drug-Resistant Small Cell Lung Carcinoma" written by Hsin-Lin Huan and  Wen Jen Lin reports on synthesis of dual peptide NPs for encapsulating two anticancer drugs against small lung carcinoma cells. Particles size of less  than 200 nm were developed and tested for their cytotoxicity and delivery showing a relevant synergistic effect. The article is interesting and suitable for Pharmaceutics, publishable after minor revision. I would reccomend few ammendmends and\or corrections. They are listed below:

1) A references literature update is suggested by adding few new relevant reviews or articles of similar NPs and showing comparable synergic effects

2) A moderate English revision\polishing by native speaker will improve comprehension of whole article and correct\revise few typos\imperefections in the text

3) A deeper discussion\elaboration of rationale behind the choice of type of cells and drugs as well as concentration is expected

4) What is time variational behaviour of MTT data? What about uptake mechanism and quantification analysis through Confocal Laser Scanning slicing ? Co-localisation study?

5) Please add visible scale bar in Figure 3C

Reviewer 2 Report

The submitted manuscript entitled: "Dual Peptide-Modified Nanoparticles Improve Combination Chemotherapy of Etoposide and siPIK3CA against Drug-Resistant Small Cell Lung Carcinoma", is a detailed study of a new approach to cancer therapy, worthwhile to be published in Pharmaceutics.

We recommend just a few changes:

In Fig3A - in the right, it is enough to restrict the y-axis to 100 counts.

In the discussion, we recommend to add: For the delivery of nanoparticle complex physical methods can also be useful  [A, B].

A. Das, M.; Solanki, A.; Joshi, A.; Devkar, R.; Seshadri, S.; Thakore, S.β-cyclodextrin based dual-responsive multifunctional nanotheranostics for cancer cell targeting and dual drug delivery, Carbohydr. Polym. 2019, 206, 694-705. doi: 10.1016/j.carbpol.2018.11.049

B. Babincová, N.; Sourivong, P.; Babinec, P.; Bergemann, C.; Babincová, M.; Durdík, S. Applications of magnetoliposomes with encapsulated doxorubicin for integrated chemotherapy and hyperthermia of rat C6 glioma.Z. Naturforsch. C 2018, 73 , 265-271. doi:10.1515/znc-2017-0110

Reviewer 3 Report

This is a very detailed in vitro validation of using dual-targeted nanoparticles to deliver siRNA targeting PIK3CA and Etoposide to drug-resistant small-cell lung cancer cell line. This is a very detailed study that is well carried out. I have some minor comments that I believe the authors can easily address.

1). There are a lot of grammatical mistakes that need to be addressed.

2). The conclusions/discussion section is very limited. Some of the results section needs to move to the discussion section. Additionally, there is a wealth of data on dual targeting using cell penetrating peptides in the context of tumor therapies, using nanoparticles, liposomes as well as dual "tandem" peptides. The results of this work need to be put in context of these prior studies.

3). Figure 2, 6 and 7A and 7B (first column) can be improved significantly by using a different format (grey, black, white and hashed) to make it easier to follow. The different patterns of hashing makes it difficult to follow the results even with a key available. Those figures can be improved and made easier to follow/understand.
